

# How to get best predictions for road monitoring using machine learning techniques

Imen Ferjani[*] and Suleiman Ali Alsaif[*]

Computer Department, Deanship of Preparatory Year and Supporting Studies, Imam Abdulrahman Bin Faisal University, Dammam, Kingdom of Saudi Arabia
[*] These authors contributed equally to this work.

## ABSTRACT

Road condition monitoring is essential for improving traffic safety and reducing accidents. Machine learning methods have recently gained prominence in the practically important task of controlling road surface quality. Several systems have been proposed using sensors, especially accelerometers present in smartphones due to their availability and low cost. However, these methods require practitioners to specify an exact set of features from all the sensors to provide more accurate results, including the time, frequency, and wavelet-domain signal features. It is important to know the effect of these features change on machine learning model performance in handling road anomalies classification tasks. Thus, we address such a problem by conducting a sensitivity analysis of three machine learning models which are Support Vector Machine, Decision Tree, and Multi-Layer Perceptron to test the effectiveness of the model by selecting features. We built a feature vector from all three axes of the sensors that boosts classification performance. Our proposed approach achieved an overall accuracy of 94% on four types of road anomalies. To allow an objective analysis of different features, we used available accelerometer datasets. Our objective is to achieve a good classification performance of road anomalies by distinguishing between significant and relatively insignificant features. Our chosen baseline machine learning models are based on their comparative simplicity and powerful empirical performance. The extensive analysis results of our study provide practical advice for practitioners wishing to select features effectively in real-world settings for road anomalies detection.

# INTRODUCTION

Monitoring the physical conditions of roads is an extremely crucial area within the transportation domain. The detection of road anomalies has been given considerable attention, and a variety of field experiments have been conducted to establish the protocol, methods, and algorithms (*Outay, Mengash & Adnan, 2020*). Road anomaly is referred to as any deviation or variation from the standard road surface conditions (*Seraj et al., 2015*). Various anomalies on the road can cause a vehicle to fall unexpectedly. Some of them are potholes, rutting, cracks, and speed bumps. This necessitates the development of automated

Corresponding authors
Imen Ferjani, eferjani@iau.edu.sa
Suleiman Ali Alsaif, saalsaif@iau.edu.sa

techniques for detecting different road anomalies. Many systems have been implemented for fast and reliable anomaly detection to prevent road accidents. Some of these systems are detecting anomalies either by using expensive and specialized road-monitoring equipment (inductive loops, video-detectors, magnetometers, etc.) or by surveying roads manually. A major drawback of these solutions is low flexibility and significant maintenance costs.

To overcome these drawbacks, a promising method for gathering real-world data is mobile sensing technology without the need to deploy special sensors and instruments (*Schuurman et al., 2012*). It is a smartphone-based method of sensing, in which data is collected using embedded sensors (*Lane et al., 2010*). Accelerometer sensors have been widely used to collect data for analysis. Based on many studies (*Astarita et al., 2012*; *Li, Chen & Chu, 2014*; *Li & Goldberg, 2018*), road roughness is a source of vibration in vehicles and a well-known cause of wear and damage to the vehicle itself, as well as to bridges and pavements. These vibrations can be effectively captured by smartphone accelerometers. There are three axes on an accelerometer (X, Y, and Z), which correspond to the longitudinal, vertical, and transverse directions of a smartphone, respectively. When acceleration is experienced in any of these axes, the accelerometer captures it (in $m/s^2$). Through analyzing these axes' signals, road anomalies can be potentially identified.

Different approaches in the literature have been proposed to classify road anomalies based on features obtained from the accelerometer sensor. Especially the machine learning algorithms which are quite diverse (*Eriksson et al., 2008*; *Perttunen et al., 2011*; *Carlos et al., 2018*; *Bridgelall & Tolliver, 2020*; *Alam et al., 2020*). In *Basavaraju et al. (2019)*; *Silva et al. (2017)*, the authors employed Multilayer Perceptron (MLP) and they made comparisons with other models such as Random Forest (RF), Support Vector Machine (SVM), and Decision Tree (DT). Other researchers used a decision tree-based classifier (*Alam et al., 2020*; *Kalim, Jeong & Ilyas, 2016*) to detect and classify different types of road anomalies. Also, Support Vector Machines have been widely used in many works (*Eriksson et al., 2008*; *Perttunen et al., 2011*; *Carlos et al., 2018*; *Bridgelall & Tolliver, 2020*; *Alam et al., 2020*).

The efficiency of any machine learning model is highly related to determining the set of features that can 'most accurately' describe the input data that has been collected from the accelerometer sensor. Several well-known features have been used in literature, such as time- and frequency-domain features. In most cases, they are mixed with other features (*Kalim, Jeong & Ilyas, 2016*; *Silva et al., 2017*; *Nunes & Mota, 2019*; *Alam et al., 2020*). There have also been reports of feature extraction based on wavelets (*Basavaraju et al., 2019*; *Brisimi et al., 2016*). It is not always clear which features will give the best classification in previous literature. Additionally, it is important to minimize the number of features to make the classifier faster and also more accurate since as more features are added, the size of the design set must also increase. Knowing which features to extract from an abundance of features in the raw data is the most challenging part. In this paper, we will go into detail about those challenges by analyzing the sensibility of some machine learning models to different categories of features. By demonstrating that specific features from accelerometer data have the greatest impact on machine learning models, we can avoid employing redundant features in the classification step as much as possible. For this

purpose, a comprehensive and up-to-date set of thirty features from the time, frequency and wavelet domains have been evaluated in this study.

Most previous studies have focused on a comparison between different machine learning models rather than investigating the sensibility of such models to selected features. Therefore, rather than studying features that practitioners should spend effort tuning, we aimed to learn which features perform better regardless of the dataset and which features are inconsequential. This paper reports the results of experiments investigating three machine learning models (Support Vector Machine, Multi-Layer Perceptron, and Decision Tree) in a large number of different configurations using two available datasets with various types of anomalies. Analyzing sensitivity, in particular, is an important part of modeling. Model builders and users are provided with useful information by emphasizing the parameters that have the most influence on the model's results. Sensitivity analysis can highlight the parameters that should be measured most precisely in order to maximize the model's accuracy. It can also provide a general assessment of the accuracy of the model. In order to develop effective design strategies, identify the parameters and interactions that have the greatest influence on the model's performance.

This paper is structured as follows: first, we present the overall machine learning-based workflow of detecting road surface anomalies from smartphone sensors, followed by a review of the most used features in the literature as well as the machine learning techniques. Then, we present our experimental results using two types of accelerometer datasets. In the following sections, the models reviewed and their limitations are discussed. Finally, we summarize the results of the study and identify challenges with detecting road surface anomalies of smartphones, along with potential research directions that should be pursued.

## METHODS AND MATERIALS

### General overview

The methodology we used to compare different feature techniques for automated detection of road anomalies is illustrated in Fig. 1. In general, online running and offline training are the two main phases of the system flow. It applies a machine learning approach to identify road anomalies based on smartphone vibrations. In the first phase, a database of annotated data is used to discover and extract useful information during the feature extraction step. A machine learning model should then be trained using the extracted features. In order to determine how the trained model functions and what its reliability is, it should be tested against new data to determine its performance. It is imperative to mention that several types of embedded sensors can be incorporated in providing input data. In our study, we will use the accelerometer sensor. The resulting trained model is used during the second phase to detect and classify anomalies from real time accelerometer readings. The feature extraction process is performed in both phases. The reason why it is a crucial step in the classification workflow.

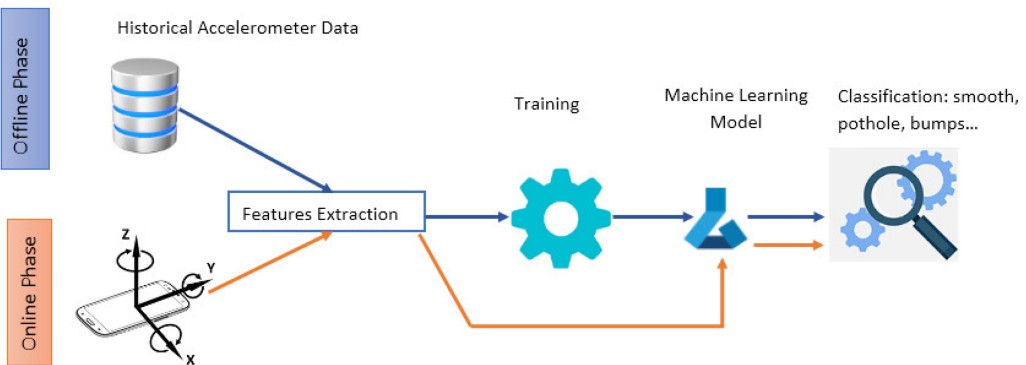

**Figure 1** **A general flow of machine learning based approach for road anomalies detection.**

## Feature extraction

The specific technique used to detect anomalies is determined by the features extracted from the accelerometer readings. Furthermore, it is crucial to extract the relevant features from accelerometer measurements, since the features become inputs for specific techniques.

### Time domain

Time domain features are often implemented as pre-processing (*Radu et al., 2018*) so they are easy to implement. They can be used to extract basic signal patterns from raw sensor data. We used a total of eight time domain features.

- Mean is the most common and easy implemented feature of the time domain. It only finds the mean of amplitude values over sample length of the signal $x_i$ that represents a sequence of $N$ discrete values $\{x_1, x_2, \ldots, x_N\}$. It is obtained using Eq. (1)

$$mean(\mu) = \frac{1}{N} \sum_{i=1}^{N} |x_i| \tag{1}$$

- Integral square (IS) uses energy of the signal as a feature. It is a summation of square values of the signal amplitude. Generally, this parameter is defined as an energy index, which can be expressed as:

$$IS = \frac{1}{N} \sum_{i=1}^{N} x_i^2 \tag{2}$$

- The variance is also most common statistical method for time domain feature extraction. It is defined as follows (Eq. (3)):

$$variance = \frac{1}{N-1} \sum_{i=1}^{N} (x_i - \mu)^2 \tag{3}$$

- The standard deviation is calculated using the following equation:

$$StandardDeviation(\sigma) = \sqrt{\frac{1}{N-1} \sum_{i=1}^{N} (x_i - \mu)^2} \tag{4}$$

- The median is the number at which the data samples are divided into two regions with equal amplitude.
- The range is the difference between maximum and minimum sample values.
- The root mean square (RMS) of a signal $x_i$ that represents a sequence of $N$ discrete values $\{x_1, x_2, \ldots, x_N\}$ is obtained using Eq. (5) and can be associated with meaningful context information.

$$x_{RMS} = \sqrt{\frac{x_1^2 + x_2^2 + \ldots + x_N^2}{N}} \tag{5}$$

- Entropy describes how much information about the data randomness is provided by a signal or event (*Shannon, 1948*).

### *Frequency domain*

Initially, the time-domain vibration signals must be transformed into frequency-domain signals using a fast Fourier transform (FFT) in order to extract the frequency-domain features (*Martens, 1992*). Our extracted features are the next:

- The Spectrum energy of the signal is equal to the squared sum of its spectral coefficients.
- Median Frequency (MF) –A frequency that divides the spectrum into two equally amplituded regions.
- Mean power (MP) –The Spectrum power average.
- Peak magnitude (PM) –The Spectrum amplitude at its maximum.
- Minimum magnitude (MM) –The lowest amplitude in the spectrum.
- Total power is defined as an aggregate of the signal power spectrum.
- The Discrete Cosine component is the first coefficient in the spectral representation of a signal and its value is often much larger than the remaining spectral coefficients.

### Time-frequency (wavelets) domain

A wavelet is a fast-decaying function with zero averaging. The nice features of wavelets are that they have local property in both spatial and frequency domain and can be used to fully represent volumes with small numbers of wavelet coefficients. With the wavelet approach.

In a time-domain analysis section, the original time-domain signal is usually decomposed into distinct bands using designed filters paired with downsampling in order to split the signal when the effective sample rate remains unchanged (*Mallat, 2009*). Based on the characteristics of the source and/or application, each produced band can be processed independently. After filtering and up-sampling, the signal is reconstructed as an approximate representation of the original. By iterating the low-pass output at each scale, the wavelet transform and its filter bank realization are repeated. Therefore, it produces a series of band-pass outputs, which are actually wavelet coefficients. As mentioned earlier, the wavelet is comprised of a high-pass filter, followed by a series of low-pass filters. For further details, see *Chau (2001)*. In our study, we used the following wavelet decomposition; the accelerometer signal is decomposed using the wavelet transform and the features defined as signal power measurements. Each of the five detailed coefficients is then summed up. At the end, it returns a total of 15 features. In Table 1, we summarize all the feature sets

**Table 1  Features summary.**

| Domain | Feature name | Symbol |
| --- | --- | --- |
| Time | Mean | μ |
| | Integral square | IS |
| | Variance | Var |
| | Standard deviation | $\sigma$ |
| | Median | 30 Med |
| | Range | Rg |
| | Root Mean Square | RMS |
| | Entropy | Ent |
| Frequency | Spectrum Energy | SE |
| | Median Frequency | MF |
| | Mean Power | MP |
| | Peak Magnitude | PM |
| | Minimum Magnitude | MM |
| | Total Power | TP |
| | Discrete Cosine | DC |
| Wavelet | Five levels - Daubechies 2 | $cD_{ij}$ |

used in our research. Our choice of features is based on a critical issue is how to combine these different sets of features in a way that may increase the performance of the model classification. We used an overlapping sliding window scheme using a window of length $w$ to group the data. The features are extracted from each window. Since the anomaly location is originally unknown and needs to be estimated, the overlapping window ensures that there exists some window that overlaps with the anomaly. Basically, the output from the previous step will be used as input to three different classifiers in order to compare their efficiency based on features selected.

## Machine learning baseline models

As popular and reliable technologies that can be applied for classifying road vibration data, SVM, Decision Trees and Neural Networks were utilized in this study.

### *Support vector machine*

SVM aims to construct a hyperplane or set of hyperplanes in an N-dimensional space (N is the number of features), which can be used for classifying the data points. Many hyperplanes could be chosen; the most optimal hyperplane is the one that has the largest margin, i.e the maximum distance between data points of both classes. Research by *Boser, Guyon & Vapnik (1992)* in 1992 revealed a method for creating nonlinear classifiers by using kernel functions (originally proposed by Aizerman, Braverman, and Rozonoer in *Aizerman, Braverman & Rozonoer, 1964*). In this algorithm, nonlinear kernel functions are often applied to transform input data into a high-dimensional feature space in which the input data become more separable compared to the original input space. Maximum-margin hyperplanes are then created. In the following text, for the purpose of this research, one

type of SVM is used, C-SVC that can incorporate different basic kernels. Given training vectors.

### Decision tree

A decision tree also known as a classification tree is a predictive model derived by recursively partitioning the feature space into subspaces that constitute a basis for prediction (_Rokach, 2016_). In the tree structures, leaves represent classifications (also referred to as labels), non-leaf nodes are features, and branches represent conjunctions of features that lead to the classifications. Decision trees are derived by doing successive binary splits (some algorithms can make multiple branches for every split). The first split will produce the most distinct data between two groups. The subgroups are then split until some stopping criteria are reached. There are differences in algorithmic approaches to establishing the distance between two groups (_Loh, 2011_).

### Multi layer perceptron

A multilayer perceptron can be used to perform classification. It is an example of a feed-forward neural network where each node performs a nonlinear activation function. The weights connecting the nodes are determined using back propagation. This network receives as input a feature vector extracted from the object to be classified. It outputs a block code, in which one high output identifies the class of the object and the other outputs are low. In order to approximate functions, only one hidden layer is required (_Cybenko, 1989_). It depends on the problem as to how many layers in the network and nodes in each hidden layer are used.

## Datasets

To validate our study, we use two different types of accelerometer measurements datasets: simulated and real data. The measurements are labeled with several categories. All the training measurements in each of these categories are split into segments according to a predetermined window size, $w$ with an overlap of $w/2$. Each segment is represented by a descriptor comprising the features mentioned previously. Two types of accelerometer datasets are used: simulated data and real data. A part of the data was used to train, while the other part was used to test. With the training data, we calibrated the parameters for each model, trying to find the values that produced the best results. To make any approach usable, it is important to reduce the amount of data it needs to be calibrated. Accordingly, in all cases, the training components are shorter than the testing components testing components.

### Simulated data

Pothole Lab (_Carlos et al., 2018_), which can be used to create virtual roads with a configurable number and nature of road anomalies, generated the first dataset we used for our experiments (DB1). Different roads were built with acceleration samples taken from the X,Y, and Z axes with a sampling rate of 50 Hz. The generated roads are divided into three types: roads without anomalies, homogeneous roads (only one kind of anomaly), and heterogeneous roads (different types of anomalies). We generated 60 virtual roads

with a total of 1591 anomalies of three types (potholes, metal speed bumps and asphalt speed bumps). Table 2 describes the details of the dataset used and the number and type of each anomaly. In this table, the term bumps represents Asphalt bumps, other terms are self-explanatory. The dataset were divided into training (70%) and testing sets (30%).

### Real data

The second dataset we employed (DB2) was published in *Gonzlez et al. (2017)* where 12 vehicles were used to perform the data collection. This dataset contains accelerometer samples from $z$-axis only, using a sampling rate of 50 Hz. More than 500 events were recorded and classified into five categories: metal bumps, worn out road, potholes, asphalt bumps, and regular road. Table 3 shows the details of the dataset used in this study as well as the distribution of anomalies used.

## Model evaluation parameters

This evaluation aims to assess the effectiveness of our models in classifying road anomalies. Following are the performance scores we calculated in this study. Accuracy, recall, and F1-score. A measure of recall is dependent on the understanding and measurement of relevance. Measures of completeness and quantity can be viewed as recall. Accuracy measures the overall performance of the classification.

- Accuracy is measured as a ratio of correctly predicted samples to the number of input samples. It gives a reasonable estimation of the model performance only if there are an equal number of samples belonging to each class. It is calculated using the following equation:

$$Accuracy = \frac{TN + TP}{TN + TP + FN + FP} \tag{6}$$

- The recall, or true positive rate, determines how well the classifier predicts positive samples. It is calculated in the following manner:

$$Recall = \frac{TP}{TP + FN} \tag{7}$$

- F1 Score is the harmonic mean between precision and recall. The greater the F1 Score, the better is the performance of our model. F1-score is calculated as the following:

$$F1 = \frac{2 \times Precision \times Recall}{Precision + Recall} \tag{8}$$

The TP (true positive) indicates that it is a valid classification of the anomaly (ground truth). A TN (true negative) indicates how many times an anomaly is classified properly as not being observed. An algorithm that falsely identifies an anomaly that was not observed is known as a false positive (FP). FN (False Negatives), is the number of cases in which an anomaly was observed (ground truth) but classified as something else by the algorithm.

Moreover, we applied cross-validation in the performance evaluation of each classification model to estimate the skill of a machine learning model on unseen data. A 10 group split is a reasonable compromise for providing robust estimates of performance and being computationally feasible. Every unique group is treated as a hold-out or test set, and all the other groups are treated as training sets. We then fit the model to the training

**Table 2  The dataset generated with Pothole Lab (*Carlos et al., 2018*).**

| Anomaly | Training set | Testing set |
| --- | --- | --- |
| Potholes | 362 | 155 |
| Metal speed bumps | 324 | 139 |
| Asphalt speed bumps | 427 | 184 |

**Table 3  The real dataset used (*Gonzlez et al., 2017*).**

| Anomaly | Training set | Testing set |
| --- | --- | --- |
| Potholes | 70 | 30 |
| Metal bumps | 70 | 30 |
| Asphalt bumps | 70 | 30 |
| Worn out road | 70 | 30 |
| Regular road | 70 | 30 |

data and evaluate it on the test data. That evaluation score is retained. As the last step, we compute the skill of the model based on the sample of model evaluation scores. However, the cross-validation accuracy is the same as the overall accuracy in all our experiments.

## RESULTS AND DISCUSSION

### Experimental setup

The experiments covered a diverse combination of factors that could potentially affect the accuracy of any machine learning model for classifying road anomalies. To determine the usefulness of these factors in the used data sets, we carried out three types of experiments:

- The effect of the accelerometer axis data. We focused on varying the input data fed to our models by applying all the possible combinations of X, Y, and $Z$-axis. We used a window size equal to 30 and a time step of 15, from which we extracted 30 features as described in Table 1.
- The effect of sliding window size: We considered multiple window sizes and we recorded the corresponding performance. We used only $Z$-axis data and all features were extracted for each window size.
- The effect of features extraction. We evaluated in our first experience the three categories of features (time, frequency, and wavelet) separately. Then, we explored all the possible combinations of features while recording the corresponding performance for each combination.

In all our experiments, we repeated the training and the testing procedures over many trials, and the average detector performance is reported. This is to discard randomness effects. The parameters in our machine learning models are set in accordance with the literature. For SVM, we used a regularization parameter equal to 1, the 'rbf' function kernel with a coefficient of $2 \times 10^{-3}$ and a tolerance of $10^{-3}$. For MLP, the 'relu' activation function is used over 300 iterations. The random state value for DT was set to zero. We used in our tests a machine having a single Nvidia Tesla K80 GPU and 12 GB of RAM.

## Analysis of accelerometer axis data

In this section, we examine how the tri-axial accelerometer sensor improves classification rate by analyzing the effectiveness and contribution of each axis. Three well-known anomalies were selected: potholes, asphalt speed bumps, and metal bumps. Experimental results were assessed using accuracy metrics. An important property of road anomalies classification models based on accelerometer data is the possibility of taking as input 3 different signals from the accelerometer axis: X, Y, and $Z$-axis. The flexibility of such architectures affords swapping in different axis data during model initialization. Therefore, we first extract all features from each window for each axis X, Y, and Z, involving time, frequency, and wavelet features. In the next step, we assess the sensitivity of the selected anomaly models according to the input axis data by depicting a combination of the different axis of the accelerometer to recognize the anomalies. Specifically, we consider all possible combinations: only $X$-axis, only $Y$-axis, only $Z$-axis, X and Y axis, X and Z axis, Y and Z axis and, all axis. We report accuracy achieved using SVM, DT, and MLP. In Table 4 the best classification rates of different anomalies against $X$-axis is 86%, $Y$-axis is 88%, $Z$-axis is 90%, X-Y axis is 89%, X-Z axis is 91%, Y-Z axis is 94%, and X-Y-Z axis is 94%.

Analyzing the relative performance of different axes shows that using the information from all axes gives the most effective results. Besides, according to the results obtained in Table 4, the use of the $Z$-axis either alone or with another axis has an impact on the performance of the models. By looking at the actual performances using all models, it shows that the top two accuracy values are 94% and 91%. When the $Z$-axis was paired with the $Y$-axis and both X-Y-axes, the highest accuracy (94%) was achieved. It is pertinent to notice that accuracy values in the last two columns of Table 4 are similar. This means that the $X$-axis has less effect on the model performance when compared to the other axis. Moreover, the $Z$-axis of the accelerometer sensor is a potential feature. The combination of the $Z$-axis with the $X$-axis increases the performance from 86% to 91%. Likewise, its combination with the $Y$-axis increases the performance from 88% to 94%. The lowest performance values reached are 80%, 81%, and 82%. In all of them, we notice the use of the $X$-axis either alone or combined with another axis. Unfortunately, simply using the $X$-axis or concatenating it with another axis does not have an improving effect. Also, combining Y and $Z$-axis data with MLP may indeed give reliable results (94%). It is imperative to note that models settings are kept the same as in the basic configuration to highlight the effect of axes used. We emphasize that MLP and SVM models gave better average results when compared to DT in most of the cases.

## Analysis of sliding window size

We reported in the previous section the performance obtained by combining various accelerometer axes. In this section, we discuss how the sliding window affects the model's performance. We first convert the accelerometer $Z$-axis signal into data windows of 30 features that overlap 50% of each other. To prevent removing relevant information from the signal, no preprocessing is applied. As long as the anomalies are diverse, this is normal practice, and even more so if the quality of the data permits it. A machine learning model is designed to identify windows when an anomaly occurs. We use a variety of window sizes

**Table 4   Accuracy (%) of our machine learning models according to accelerometer axes used.**

| Model | X-axis | Y-axis | Z-axis | (X-Y) axis | (X-Z) axis | (Y-Z) axis | (X-Y-Z) axis |
|-------|--------|--------|--------|------------|------------|------------|--------------|
| SVM | **86**% | **88**% | 87% | **89**% | **91**% | 93% | 93% |
| DT | 81% | 83% | 87% | 83% | 80% | 93% | **94**% |
| MLP | 84% | 85% | **90**% | 82% | 85% | **94**% | 93% |

for evaluation, ranging from 10 up to 100 in steps of 10. Previous anomalies detection systems have mostly used this interval. Figure 2 shows the performance results for diverse window sizes and models. Since we only need to see the trend in model accuracy as the window size is altered (rather than the absolute performance of each model), the accuracy is only shown as a percentage change as the sliding window size is increased.

Experimental results have shown that window sizes differ between models. One reason for this may be that different anomalies have varying durations. In DB1, SVM and DT models show an increasing performance with larger windows. For size 5, a minimum performance of 81% is achieved, which nevertheless increases up to 94% when the window is enlarged to 80. For some window sizes, the performance improves by less than 5% as compared to the performance at size 80. Actually, from a window size greater than 80, no significant benefits are obtained in all models' performances. Regarding DB2, different performance values have been recorded varying between 25% and 59%. Conversely to DB1, all models, especially DT and MLP, suffer from a worsening of performance when the window size is increased beyond 50. Window sizes between 45 and 50 provide the best performance for DT. This technique provides the highest level of performance, with a 94% accuracy rate. Upper and lower values of 30 and 60 generally decrease the classifier's performance. The MLP model is showing an oscillating behavior when increasing the window size.

It was shown that each model has its optimal window size based on the results. However, considering the figure in Fig. 2, a reasonable range of size might be around 30 to 60 for our used datasets comprising three types of anomalies for DB1 and four types in DB2. In addition, the classification accuracy trend seems to be different for each model in DB2. Additionally, it is also found that the window size is highly related to the type of anomaly. Studying each anomaly, in particular, would be of interest.

The choice of small window size is a challenging task when using machine learning techniques because the cost of labeling every short interval of data is extremely high. Several approaches may be used for solving such a problem, including incremental learning or reinforcement learning, which can maintain and modify the expert model over time without the need to re-train it. The main limitation that should be noted, which should be studied further is that only a few benchmarking datasets are available for the classification of road anomalies.

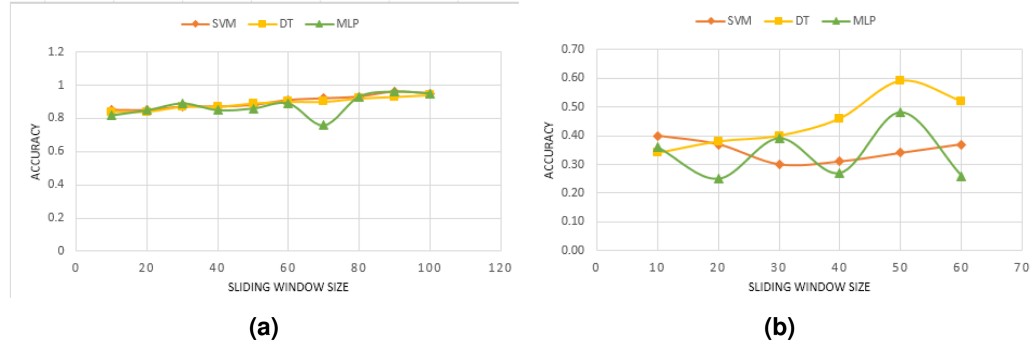

**Figure 2** Effect of the overlapping window size using only *Z*-axis for: (A) DB1, (B) DB2.

**Table 5** Performance evaluation of our machine learning models according to domain feature used (without domain combination).

|  | Model | Simulated data | | | | Real data | | | |
|---|---|---|---|---|---|---|---|---|---|
|  |  | Accuracy | Precision | Recall | F1 | Accuracy | Precision | Recall | F1 |
| Time | SVM | **0.88** | 0.77 | 0.30 | 0.33 | 0.43 | 0.38 | 0.32 | 0.28 |
|  | DT | 0.87 | 0.56 | 0.55 | 0.55 | 0.36 | 0.33 | 0.33 | 0.33 |
|  | MLP | **0.88** | 0.43 | 0.35 | 0.38 | **0.47** | 0.42 | 0.37 | 0.35 |
| Frequency | SVM | **0.86** | 0.74 | 0.26 | 0.26 | 0.29 | 0.21 | 0.20 | 0.16 |
|  | DT | 0.83 | 0.42 | 0.42 | 0.42 | **0.34** | 0.30 | 0.30 | 0.30 |
|  | MLP | **0.86** | 0.27 | 0.25 | 0.24 | 0.24 | 0.04 | 0.2 | 0.07 |
| Wavelet | SVM | 0.89 | 0.59 | 0.39 | 0.41 | 0.48 | 0.46 | 0.38 | 0.36 |
|  | DT | 0.85 | 0.49 | 0.51 | 0.49 | 0.38 | 0.34 | 0.34 | 0.34 |
|  | MLP | **0.90** | 0.69 | 0.42 | 0.45 | **0.49** | 044 | 0.39 | 0.37 |

## Analysis of features

In this experiment, the basic configuration is held constant, while we only alter the number of features extracted for each model. Each of the three feature domains–Time, Frequency, and Wavelet–is considered separately. We report results in Table 5.

When comparing the wavelet with the other feature domains, it provides a very competitive level of accuracy. An MLP model achieved an accuracy of 90% in DB1 and 49% in DB2 (about one percent greater than an SVM model). Similar results were obtained with MLP, which achieved the highest accuracy when using wavelet decomposition. Theoretically, MLP classifiers implement empirical risk minimization, whereas SVMs minimize structural risk. So, both MLPs and SVMs are efficient and generate the highest classification accuracy for our used datasets. However, from our results, we have noticed that the DT achieved the lowest accuracy for both DB1 and DB2. A possible explanation is that decision trees work better with training data which does not exist in our datasets. (DB1 contains four categories and DB2 contains five categories).

Another key result we obtained is that the model accuracy is higher when using a simulated dataset. Models using data from road simulators (where the anomalies detected are recorded in controlled conditions) lack information about many factors that can affect

**Table 6  Performance evaluation of our machine learning models according to domain feature used (with domain combination).**

|  | Model | Simulated data | | | | Real data | | | |
|---|---|---|---|---|---|---|---|---|---|
|  |  | Accuracy | Precision | Recall | F1 | Accuracy | Precision | Recall | F1 |
| T + F | SVM | 0.86 | 0.75 | 0.26 | 0.26 | 0.29 | 0.21 | 0.20 | 0.16 |
|  | DT | 0.86 | 0.51 | 0.51 | 0.51 | 0.36 | 0.33 | 0.33 | 0.33 |
|  | MLP | 0.86 | 0.33 | 0.25 | 0.24 | 0.24 | 0.04 | 0.2 | 0.07 |
| T + W | SVM | 0.89 | 0.59 | 0.39 | 0.41 | 0.49 | 0.48 | 0.40 | 0.38 |
|  | DT | 0.88 | 0.55 | 0.57 | 0.56 | 0.42 | 0.39 | 0.39 | 0.39 |
|  | MLP | **0.90** | 0.67 | 0.42 | 0.47 | **0.52** | 0.54 | 0.44 | 0.44 |
| F + W | SVM | 0.87 | 0.94 | 0.28 | 0.30 | 0.30 | 0.22 | 0.20 | 0.14 |
|  | DT | 0.86 | 0.49 | 0.50 | 0.49 | 0.40 | 0.36 | 0.35 | 0.35 |
|  | MLP | 0.87 | 0.54 | 0.27 | 0.27 | 0.34 | 0.34 | 0.28 | 0.21 |
| T + F + W | SVM | 0.87 | 0.94 | 0.28 | 0.30 | 0.30 | 0.22 | 0.20 | 0.14 |
|  | DT | 0.87 | 0.53 | 0.56 | 0.54 | 0.42 | 0.39 | 0.39 | 0.39 |
|  | MLP | 0.87 | 0.57 | 0.47 | 0.47 | 0.42 | 0.30 | 0.30 | 0.25 |

the signal accuracy such as driver behavior. On the other hand, the accuracy obtained from real data seems to give a reliable evaluation of any machine learning model.

A combination of several features has also been considered. Table 6 presents the obtained results. With the application of time and wavelet features, the accuracy increases significantly with an accuracy of 90% for DB1 and 52% for DB2. However, by using all the features, the accuracy is not improved, and the overall performance is lower.

The confusion matrices of the highest accuracy obtained using wavelet and time features are shown in Fig. 3. In DB1, the pothole is sometimes confused with no anomaly. However, there is almost no confusion between the metal bump and other anomalies. As a metal bump will have a different effect from a pothole or regular road, it makes sense because a metal bump is expected to have a distinctive effect.

The results are different in DB2 as we notice that the distribution of accuracy across categories is nearly the same. One possible reason is that DB2 is a collection of more realistic data.

The source code for these experiments has been released to researchers on request so they can replicate this work and examine more features and models. This study was conducted to encourage additional research exploring more features and models. This research will also be useful in many other applications, such as activity recognition, to select the appropriate feature extraction techniques. This study describes the methodological challenges of extracting features from various machine learning models input and explores what features are suitable for analyzing road anomalies. In addition, deep learning models may be compared to standard machine learning models.

To ensure excellent accuracy, other techniques could also be incorporated in addition to the well-known ones used in this study. Several feature techniques combined, for instance, could produce acceptable results. Many of these techniques, however, tend

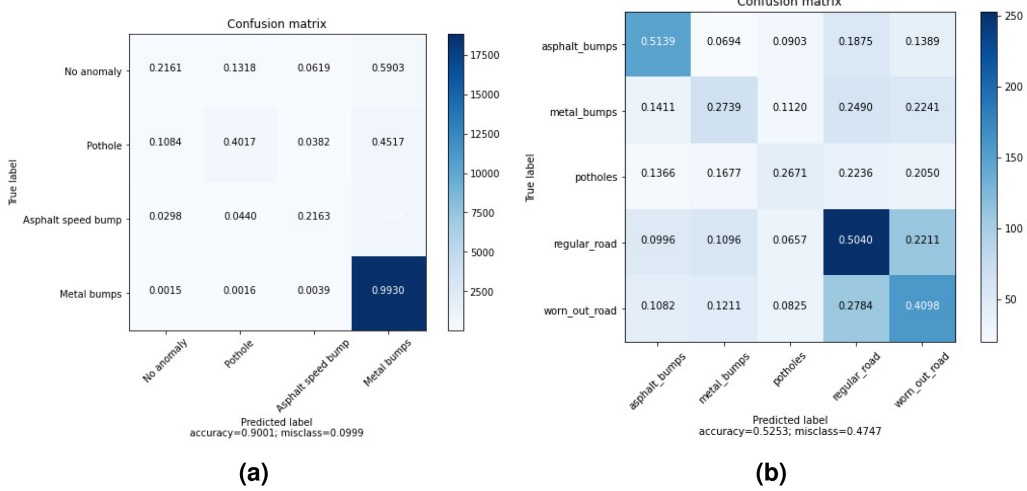

**Figure 3** Averaged Confusion matrices of MLP with wavelet and time features (using only $Z$-axis) for: (A) DB1, (B) DB2.

to be computationally expensive, and may not be suitable for applications requiring near-real-time processing.

The 'most appropriate' domain of features depends on the model. It would seem, however, that using the feature domains separately would yield the lowest level of precision. Another salient practical point is that the time required to extract features varies according to the domain. In practice, these findings suggest that it may be beneficial to use different domain-specific features to improve classification performance.

## Comparison with literature works

To evaluate the ability of our approach to competitively produce accurate predictions of different types of anomalies, we made a comparison with different literature works detailed previously. It is imperative to note that the datasets used in these methods are different. Thereby, an exact comparison is not possible due to different data sets and test setups. We mention in Table 7 the types of anomalies detected in each method as well as the accuracy of the predictive model applied. The accuracy values reported in the table are directly cited from the original publications. Using our proposed set of features, we achieved an accuracy average of 94%, which is a very competitive result compared to state-of-the-art works. Previous works in literature considered that only the $Z$-axis could represent the anomaly information. Also, their efforts have been focused on threshold heuristics. However, these strategies have shown their limitations when implemented in real-world applications. Remarkably, achieving a high classification performance is not related to the number of features used. A well-chosen set of features may result in a much more efficient classifier. This last point implies that the choice of discriminators plays an imperative role in detecting road irregularities.

**Table 7  Accuracy comparison between the best classifier reported in this work and works from the literature.**

| Reference | Anomalies detected | Technique | Accuracy average |
|---|---|---|---|
| *Eriksson et al. (2008)* | Potholes | Threshold | 92.4% |
| *Fazeen et al. (2012)* | Pothole, Bumps, Rough, Smooth uneven | Threshold | 85.6% |
| *Martinez, Gonzalez & Carlos (2014)* | potholes, speed bumps, metal humps, rough roads | ANN, Logistic regression | 86% |
| *Gonzlez et al. (2017)* | Potholes, Metal bumps, Asphalt bumps, Regular road, Worn out road | ANN, SVM, DT, RF, NB, KR, KNN | 93.8% |
| *Alam et al. (2020)* | speed-breakers, potholes, broken road patches | Decision tree | 93% |
| **This work** | Potholes, Metal speed bumps, Asphalt speed bumps | MLP, DT, SVM | **94%** |

## CONCLUSION AND FUTURE WORK

This work tackles the problem of classifying road anomalies using machine learning techniques. An extensive experiment was conducted to investigate three machine learning models for classifying road anomalies. In conclusion, we summarize our main findings and conclude with concrete advice for researchers and practitioners wishing to apply these machine learning models to real-world road assessment systems. The results show that the accuracy rates of machine learning models trained with features extracted from all three coordinate axes are significantly higher than those trained with the axis perpendicular to the ground only ($Z$-axis). All three machine learning techniques explored in this paper show this trend. The results support our hypothesis that all three axes of the coordinate system provide useful information about the road's condition. Based on the results of a sensitivity analysis, an overlapping window strategy with a size between 30 and 60 was selected for better performance. Further analysis is needed to completely understand the relation between the window size and the type of anomaly specified. Another important finding of our study is the sensibility of machine learning models to the selected features. We built a better feature vector that increases classification performance. This vector is based on wavelet features that outperform other domain features. Also, the MLP model has a reasonably high level of accuracy in classifying anomalies. With the extraction of all domain features, an overall success rate of 94% is observed when compared to the ground truth. From this point of view, merging different feature domains, especially wavelet features, seems to be more effective for preserving most road anomaly characteristics. However, separating each domain seems to be inefficient. This article discusses only datasets that contain roads with characteristics similar to those considered for this work. Our findings here may not apply to all cases. Nevertheless, we believe these suggestions are likely to be useful for researchers interested in integrating machine learning approaches into real-world anomaly detection tasks.

### Funding
The authors received no funding for this work.

### Competing Interests
The authors declare there are no competing interests.

### Author Contributions
- Imen Ferjani conceived and designed the experiments, performed the experiments, analyzed the data, performed the computation work, prepared figures and/or tables, authored or reviewed drafts of the paper, and approved the final draft.
- Suleiman Ali Alsaif conceived and designed the experiments, performed the computation work, authored or reviewed drafts of the paper, and approved the final draft.

### Data Availability
The data is available at the following sites:

- M. R. Carlos, M. E. Aragón, L. C. González, H. J. Escalante and F. Martínez, "Evaluation of Detection Approaches for Road Anomalies Based on Accelerometer Readings–Addressing Who's Who," in IEEE Transactions on Intelligent Transportation Systems, 19(10), 3334-3343. doi: https://dx.doi.org/10.1109/TITS.2017.2773084

- L. C. González, R. Moreno, H. J. Escalante, F. Martínez and M. R. Carlos, "Learning Roadway Surface Disruption Patterns Using the Bag of Words Representation," in IEEE Transactions on Intelligent Transportation Systems, vol. 18, no. 11, pp. 2916-2928, Nov. 2017. doi: https://dx.doi.org/10.1109/TITS.2017.2662483

link for used datasets: https://www.accelerometer.xyz/datasets/

The code is available at GitHub:

https://github.com/imenFerjani/Road-Anomalies-Detection/blob/05bca80bd51e5017d3362f4e0592b6c2c6e6f627/Road_Anomalies_Detection_Project.ipynb.

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
