# Peer review of "How to get best predictions for road monitoring using machine learning techniques"

_PeerJ Computer Science, doi:10.7717/peerj-cs.941_

## Round 0.1 · original submission · Minor Revisions

Dear Authors,

Good work but there are a few points which must be addressed to improve the quality of your article.

The experimental results must be analyzed - a comparison in table format could be a useful addition.

The results should be clearly summarized.

Remove grammatical and spelling errors from the analysis section and introduction section.

Along with the above please address the reviewers' comments in your Minor Revision.

Thanks
Prof. M. Nageswara Rao
Academic Editor, PeerJ Computer Science

Reviewer 1 ·

Basic reporting

The importance of the application domain should be shortly mentioned and emphasized in the abstract.

The study being retrospective and no independent assessor for the outcome would carry great bias in different issues including selection and exclusion of patients and outcome assessment.

Experimental design

In each part, in the experimental results and analysis section, before giving the driven conclusions, the results should be summarized, explaining what is given in Table, column, row, etc. This is necessary, so that the reader can validate the outcomes obtained.

Include more technical discussions on the observations would strengthen the paper's contribution.

Validity of the findings

The comparison is not fair to verify the proposed method. Include more technical discussions on the observations would strengthen the paper's contribution.
More recent references should also be included: Only 3 out of 25 cited papers are published in the last 5 years.
Language, grammar errors need to be addressed and add latest references.

Additional comments

The study being retrospective and no independent assessor for the outcome would carry great bias in different issues including selection and exclusion of patients and outcome assessment.

Reviewer 2 ·

Basic reporting

1. extracting 30 features what are those and not explained properelly

Experimental design

architecture is not available. if possible add it

Validity of the findings

good

Additional comments

NO

Reviewer 3 ·

Basic reporting

The article flows well. However, there are several grammatical and spelling errors in the submission. Please ensure you correct them.

Experimental design

The datasets used in the paper should be further defined so that the reader understands different nuances involved in it. In the current state, the paper only describes the datasets in couple of sentences.

Validity of the findings

I am fine with the methods used. Please include the majority class baseline for your models in table 4.

---

## Round 0.2 · Minor Revisions

Good work with the revisions. Just a few more changed are required, particularly for language, clarity and motivations.

Reviewer 2 ·

Basic reporting

1. Found a few Spelling Mistakes in the paper
2. literature is not there
3. the Problem: If there was no problem, there would be no reason for writing
a manuscript, and definitely no reason for reading it. So, please tell
readers why they should proceed with reading. Experience shows that for this part
a few lines are often sufficient.

Experimental design

ok

Validity of the findings

good

Reviewer 3 ·

Basic reporting

Authors have addressed all my concerns from my previous review

Experimental design

Authors have addressed all my concerns from my previous review

Validity of the findings

Authors have addressed all my concerns from my previous review

---

## Round 0.3 · accepted · Accept

Good and timely work. The revisions addressed all comments, as per the suggestions.

Please keep up the good work.

Reviewer 2 ·

Basic reporting

ok

Experimental design

ok

Validity of the findings

ok

Additional comments

Modified as per the comments